# R-Sparse: Rank-Aware Activation Sparsity for Efficient LLM Inference

**Zhenyu Zhang[1], Zechun Liu[2], Yuandong Tian[2], Harshit Khaitan[2], Zhangyang Wang[1], Steven Li[2]**
[1]The University of Texas at Austin, [2]Meta AI
`zhenyu.zhang@utexas.edu`, `stevenlx@meta.com`

## Abstract

Large Language Models (LLMs), while demonstrating remarkable capabilities across various applications, present significant challenges during inference due to their substantial model size, especially when deployed on edge devices. Activation sparsity offers a promising solution to reduce computation and memory movement, enabling more efficient inference, particularly for small-batch on-device applications. However, current approaches face limitations with non-ReLU activation function, which are foundational to most advanced LLMs, or require heavy continual training. Additionally, the difficulty in predicting active channels and limited achievable sparsity ratios constrain the effectiveness of activation sparsity-based methods. In this paper, we introduce `R-Sparse`, a training-free activation sparsity approach capable of achieving high sparsity levels in advanced LLMs. We conducted two preliminary investigations into how different components contribute to the output within a single linear layer and found two key observations: (i) the non-sparse components of the input function can be regarded as a few bias terms, and (ii) The full computation can be effectively approximated by an appropriate combination of input channels and weight singular values. Building on this, we replace the linear layers in LLMs with a rank-aware sparse inference method that leverages the sparsity of input channels and singular value components, eliminating the need for active channel prediction like the output sparsity based approaches. Experiments on Llama-2/3 and Mistral models across ten diverse tasks demonstrate that `R-Sparse` achieves comparable performance at 50% model-level sparsity, resulting in a significant 43% end-to-end efficient improvements with customized kernels. The code is available at `https://github.com/VITA-Group/R-Sparse`.

## 1 Introduction

Large Language Models (LLMs) have become ubiquitous due to their remarkable capabilities, powering applications from virtual assistants to automated content creation. However, their impressive performance comes with significant computational and memory costs due to their enormous parameter counts. This poses significant challenges for latency-sensitive applications, particularly for deployments on edge devices. To address this, network pruning or sparsity (Frantar & Alistarh, 2023; Sun et al., 2023; Yin et al., 2023; Ma et al., 2023) is an effective solution. These strategies operate in a data-independent manner with different levels of pruning granularity, *e.g.*, unstructured, semi-structured, or structured. While more structured pruning approaches leads to more limited sparsity levels, unstructured sparsity introduces greater challenges for efficient hardware implementation.

Recently, activation sparsity (Liu et al., 2023; Mirzadeh et al., 2023; Dong et al., 2024; Lee et al., 2024) has emerged as a promising solution that dynamically loads only the active channels and their corresponding weight rows or columns from off-chip HBM (NVIDIA, 2020) to on-chip SRAM, significantly alleviate the latency and memory cost when equipped with optimized system implementations (Song et al., 2023). Designing activation sparsity functions in a structured, data-dependent way, can make the specified network more hardware-friendly while also achieving higher sparsity levels compared to traditional pruning techniques.

Despite the promising progress, several challenges remain: (i) *Feasibility for non-ReLU based LLMs*: ReLU eliminates the negative part of activations, enabling a lossless approximation when skipping

the computation of corresponding channels (Liu et al., 2023). However, most advanced LLMs now use non-ReLU activations like SiLU (Elfwing et al., 2018) and GELU (Hendrycks & Gimpel, 2016), which retain small negative values, requiring extensive continual pre-training to obtain meaningful activation sparsity (Song et al., 2024a; Zhang et al., 2024a; Mirzadeh et al., 2023; Song et al., 2024b). Such training process can involve up to 150B tokens, taking approximately one month on 64 A100 GPUs. (ii) *Difficulty in Predicting Active Channels*: Previous approaches identify critical channels within the hidden activations of MLP blocks, facing significant challenges in predicting the active channels before performing the computation. Common strategies include exploiting the similarity of activated channels across semantically similar tokens (Dong et al., 2024), leveraging the activations after the gate projection (Lee et al., 2024), or using a learnable predictor (Liu et al., 2023), while the accuracy of active channel prediction will highly affect their effectiveness. (iii) *Limited Sparsity Levels*: For approaches that do not rely on extensive retraining (Lee et al., 2024; Dong et al., 2024), only 50% sparsity within MLP blocks can be achieved, leading to a model-level sparsity of one third. Achieving higher levels of overall sparsity remains a significant challenge.

This paper targets a training-free activation sparsity approach that is: (i) feasible for non-ReLU based LLMs; (ii) unaffected by the difficulty of predicting active channels; and (iii) capable of achieving higher sparsity levels. While previous methods focus on output activation sparsity (Liu et al., 2023; Lee et al., 2024), requiring prior prediction of important channels, our approach leverages input activation sparsity, identifying active channels directly from the input without the need of prediction. Furthermore, recent studies (Mirzadeh et al., 2023; Song et al., 2024a;b) have shown that directly removing the non-sparse components only achieves limited sparsity while with extensive training, sparsity ratios can be pushed to as high as 90%. This sparsity gap raises an natural question: Is the non-sparse portion of the activation truly necessary for maintaining model performance, or can we employ a lightweight strategy to mitigate the non-sparse part without resorting to heavy pre-training? Motivated by this, we apply a multi-phase ReLU function to the non-sparse channels, the corresponding activations will then be rounded to a few discrete values. As the number of discrete values increases from 1 to 2, performance can be significantly improved, even at a sparsity level of 90%. The output components associated with the non-sparse portion can then be approximated by a few bias terms, indicating a low-rank structure for these components.

To better understand the low-rank structure, we analyze the importance of each input channel in the activations and each singular value component of the weights, to the output activations. As shown in Figure 1, we observe a highly sparse structure where an appropriate combination of input channels (green rectangle) and singular value components (yellow rectangle) can effectively approximate the full computation. Building on these, we propose R-Sparse, a simple yet effective framework that decompose the computation of each linear layer with a sparse and low-rank components. For the sparse portion, our approach identifies sparse channels by selecting those with large magnitude values and loads only the corresponding rows of weights into SRAM for computation. For the low-rank components, we route the non-sparse channels to a low-rank modules that obtained from an offline low-rank decomposition of the original weights. R-Sparse can be applied to both attention and MLP modules that achieves higher sparsity levels. Additionally, we find the patterns of sparse and low-rank combinations vary across different layers. With that, we employ an evolutionary search algorithm to identify the optimal ratios for the sparse components in each layer within LLMs, resulting in enhanced performance.

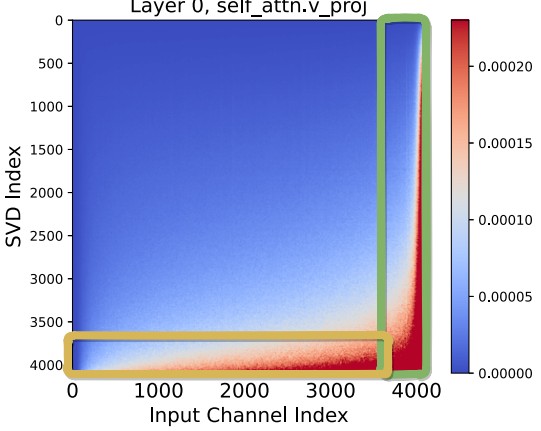

Figure 1: Contributions of each input channel and singular value components. The measurement metric is detailed in Section 3.3. Results are obtained from Llama-2-7B with 16 training samples from C4. Both the input channel and SVD components are sorted from small to large for better visualization.

We conduct extensive experiments on three representative LLM families: Llama-2 (Touvron et al., 2023), Llama-3 (Dubey et al., 2024), and Mistral (Jiang et al., 2023), across ten tasks, including common-sense reasoning, language modeling, and text summarization. Our approach achieves 50% model-level sparsity while maintaining performance comparable to the full model. Additionally, by utilizing a customized kernel, we demonstrate up to 43% end-to-end improvements in generation speed. Furthermore, `R-Sparse` is compatible with weight quantization for further efficiency gains.

## 2 RELATED WORKS

### 2.1 EFFICIENT LLM INFERENCE

The inference process of LLMs is typically memory-intensive due to the large number of parameters and the huge KV cache required to store intermediate key and value embeddings. To reduce memory overhead, various strategies have been investigated, including removing redundant components through pruning or sparsification (Frantar & Alistarh, 2023; Sun et al., 2023; Yin et al., 2023; Ma et al., 2023; Zhang et al., 2024b; Xiao et al., 2023b; Jiang et al., 2024); quantizing data into lower bit formats (Frantar et al., 2022; Lin et al., 2024; Xiao et al., 2023a; Chee et al., 2024; Kim et al., 2023; Egiazarian et al., 2024; Liu et al., 2024b); and distilling large models into smaller or more efficient architectures (Bick et al., 2024; Hinton, 2015; Sreenivas et al., 2024). Additionally, some approaches focus on developing efficient architectures (Gu & Dao, 2023; Peng et al., 2023; Yang et al., 2023) or optimizing hardware (Dao et al., 2022; Kwon et al., 2023; Alizadeh et al., 2023), enhancing the efficiency of LLM inference and making them more accessible on edge devices. This work focuses on mitigating the overhead from the large model sizes while compression techniques for KV cache are orthogonal to weight reduction and can be naturally combined that we will explore in the future.

### 2.2 ACTIVATION SPARSITY

Several studies have demonstrated that activations within the MLP blocks of transformers are highly sparse (Geva et al., 2020; Li et al., 2022; Dettmers et al., 2022). This sparsity primarily arises from ReLU activations, where negative values are zeroed out, providing a natural, lossless opportunity for accelerating inference in LLMs like OPT (Zhang et al., 2022). However, most modern LLMs use activation functions like SiLU or GeLU, which retain small negative values. Directly replacing with the ReLU activation would impair model functionality. To address this challenge, a common strategy is "ReLUfication" where the original activations are replaced with ReLU, followed by extensive continual training to recover performance (Zhang et al., 2024a; Mirzadeh et al., 2023; Song et al., 2024b;a). However, this approach introduces significant computational overhead, limiting its accessibility. Recent training-free methods (Lee et al., 2024; Dong et al., 2024) have made progress in applying sparsity to non-ReLU models, achieving modest sparsity ratios (e.g., 50% in MLP blocks and up to 33% model-wide). Additionally, most previous works focus on the sparse structure of output activations, requiring extra effort to identify active channels before computation (Dong et al., 2024; Liu et al., 2023; Lee et al., 2024), with the accuracy of channel prediction significantly affecting performance. In our work, we shift the focus to the sparse structure of input channels and singular value components, eliminating the need for active channel prediction while feasible for both attention and MLP blocks, leading to higher sparse ratios without additional training. One concurrent work (Liu et al., 2024a) shares a similar intuition but focuses solely on input channels that can be viewed as a special case of our framework.

## 3 METHODOLOGY

This section starts from a brief overview of LLM inference and the notations used throughout the paper. Following this, we present two interesting observations: (I) the contribution of non-sparse (*i.e.*, small-magnitude) input channels can be converted into biases, and (II) the full computation can be effectively approximated with an appropriate combination of input channels and singular value components. Motivated by these, we detail our proposed inference framework `R-Sparse`, along with the evolutionary search algorithm for determining the optimal sparsity recipe.

### 3.1 PRELIMINARY

LLM inference typically consists of two stages: ❶ the pre-filling stage, where a batch of prompts containing multiple tokens is processed by the model, and ❷ the decoding stage, where new tokens are generated incrementally. The decoding phase is often memory-bounded, and its iterative mechanism amplifies the overhead associated with loading parameters into on-chip memory, becoming the main bottleneck during inference. However, activation sparsity mitigates this by enabling the selective loading of only active rows or columns of the weights into SRAM at each decoding stage. In the following, we focus primarily on the decoding phase.

Consider a typical LLM architecture, where each block contains seven linear layers. The attention part comprises four matrices: $\mathbf{W}_q, \mathbf{W}_k, \mathbf{W}_v, \mathbf{W}_o \in \mathbb{R}^{n \times n}$, while the widely used MLP block Touvron et al. (2023); Dubey et al. (2024) includes three matrices: $\mathbf{W}_{up}, \mathbf{W}_{gate} \in \mathbb{R}^{m \times n}$ and $\mathbf{W}_{down} \in \mathbb{R}^{n \times m}$ ($n$ and $m$ stands for the dimension of model embedding and hidden activations within MLP blocks, respectively). The computational process of the MLP block can be formulated as $Y = H\mathbf{W}_{down}^T$, where $H = X\mathbf{W}_{up}^T \odot \sigma(X\mathbf{W}_{gate}^T)$.

### 3.2 MOTIVATION CASE I: NON-SPARSE COMPONENTS ARE BIASES

We first carry out a preliminary investigation into how sparsification of input activations influences the final performance. We use a soft multi-phase ReLU function $\sigma_{\mathcal{T}}(\cdot)$ to approximate the non-ReLU activation functions $\sigma(\cdot)$, which is defined as:

$$\sigma_T(x) = \begin{cases} x & \text{if } x \geq T_0 \\ \frac{T_i + T_{i+1}}{2} & \text{if } T_{i+1} \leq x < T_i \end{cases}$$

where $\mathcal{T} = \{T_0, T_1, .., T_{l-1}\}$ and $l$ determines the softness of the sparsification operation. When $T_0 = 0$ and $l = 1$, this is equivalent to standard activation sparsity achieved by ReLU where all non-sparse part ($x < 0$) are masking out as zero. Note that we define the sparse components as the values that remain unchanged after the activation function, while the sparsity ratio is measured as the proportion of values that being changed. Additionally, we set $T_{l-1}$ as the minimum value of input and the sparsity is defined as the ratios of $x < T_0$. As shown in Figure 2. By simply increasing $l$ from 1 to 2, the degraded performance can be easily recovered, even at a sparsity ratio of

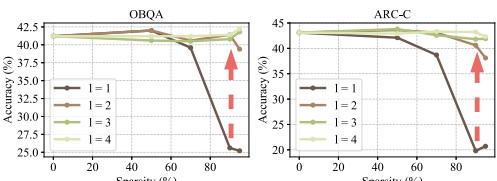

Figure 2: Accuracy of Llama-2-7B on OpenBookQA (Mihaylov et al., 2018a) (OBQA) and ARC Challenge (Clark et al., 2018a) (ARC-C) tasks.

90%. Additionally, we use $\mathcal{U}_i$ to represent the subset of channels in $H$ that satisfy $T_{i+1} \leq H_k < T_i$ ($k \in \mathcal{U}_i$). The corresponding output $Y$ can then be decomposed into the sparse part $Y_s$, where $H_k \geq T_0$, and the residual part $Y_r$, as:

$$Y_r = \sum_{j=0}^{l-2} \frac{T_j + T_{j+1}}{2} \left( \sum_{k \in \mathcal{U}_j} \mathbf{W}_{down}^T[:, k] \right)$$

The subset of channels $\mathcal{U}_j$ is input-dependent and each term $\sum_{k \in \mathcal{U}_j} \mathbf{W}_{down}^T[:, k]$ can be viewed as a data-dependent bias $B_j$. This allows the non-sparse components to be effectively approximated with a few biases. We will show later how these data-dependent biases can be converted into static biases and being pre-computed. With only two biases, the sparsity ratio is significantly increased to 90%.

### 3.3 MOTIVATION CASE II: RANK-AWARE ACTIVATION SPARSITY

Although it's costly to obtain the input-dependent biases on the fly. we observe that the space spanned by the biases across thousands of tokens exhibits a low-rank structure, *e.g.*, for each token $i$, we use two biases to approximate the residual part $Y_r^i = B_0^i + B_1^i$. By concatenating 4000 biases from 2000 tokens, we obtain a bias matrix $\mathbf{M}$, where $\mathbf{M}[:, 2i] = B_0^i$ and $\mathbf{M}[:, 2i + 1] = B_1^i$

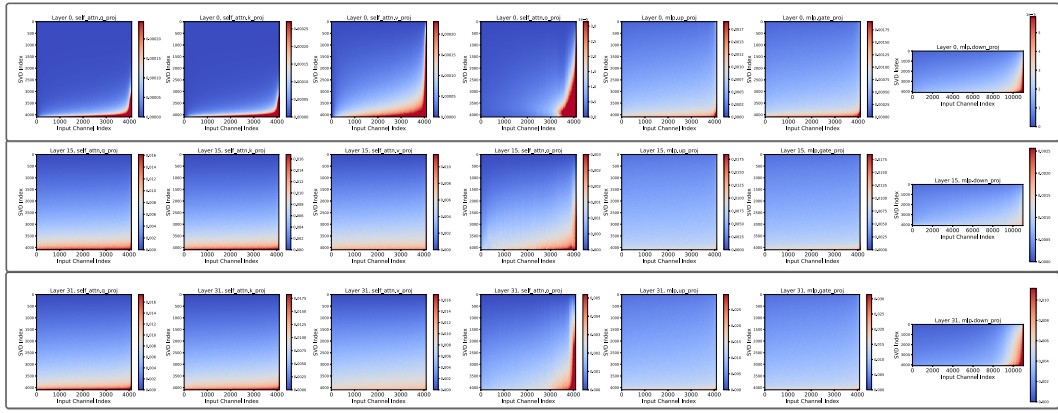

Figure 3: Importance of each input channel and singular value. Zoom in for better visualization. Results are obtained with the pretrained Llama-2-7B model and 16 samples from the C4 training dataset, each with a sequence length of 4096. Each subfigure corresponds to the results of different layers, with the horizontal axis representing the input channel index and the vertical axis representing the singular value index. The top, middle, and bottom subfigures represent the results of the first, middle, and last layers, respectively.

($i \in \{1, 2, ..., 2000\}$). We find the stable rank of $\mathbf{M}$ is approximately 400. Inspired by this, we further explore the relationship between weight SVD components and sparse activations. Given a pre-trained linear layer $Y = X\mathbf{W}^T$, $\mathbf{W} \in \mathbb{R}^{n \times m}(n \leq m)$, we perform singular value decomposition (SVD) on the weight matrix, that $\mathbf{W} = \mathbf{U}\Sigma\mathbf{V}^T = \sum_{i=1}^{n} \sigma_i \mathbf{U}[:, i]\mathbf{V}^T[:, i]$. The output $Y$ can then be expressed as $Y = \sum_{i=1}^{n} \sum_{j=1}^{m} \sigma_i X_j \mathbf{V}[j, i]\mathbf{U}^T[:, i]$ where $\mathbf{S}_{i,j} := \sigma_i X_j \mathbf{V}[j, i]$ measures the contribution of the $j$-th input channel and the $i$-th SVD components. We collected the distribution of $\mathbf{S}$ for Llama-2-7B (Touvron et al., 2023) using 16 training samples from the C4 dataset (Dodge et al., 2021), each containing 4096 tokens. For better visualization, both the rows and columns of $\mathbf{S}$ were sorted independently. Across different linear layers in either Attention or MLP blocks, the primary contributions are concentrated in the lower-right corner. Additionally, almost all layers exhibit significant sparse property, although some variation exists across layer types and blocks. For instance, the *o.proj* layer exhibits a greater reliance on smaller singular values compared to the *q.proj* and *k.proj* layers. This observation also aligns with with recent studies (Jaiswal et al., 2024), which demonstrate that *q.proj* and *k.proj* can be more easily compressed via low-rank approximation. Moreover, middle layers tend to display higher sparsity, while initial and final layers are more difficult to be sparsified, aligning with the general experience that the beginning and final layers of LLMs are harder to be compressed (Yin et al., 2023).

### 3.4 R-SPARSE

Building on the observation of rank-aware activation sparsity, we propose the R-Sparse inference framework. An overview of R-Sparse and its comparison with other techniques is presented in Figure 4. For a given score matrix $\mathbf{S}$, previous methods that based on activation sparsity typically remove the left portion of $\mathbf{S}$, while low-rank compression techniques eliminate the upper portion. However, since the most significant components concentrate in the bottom-right area, an ideal approach would be to remove the top-left part. To efficiently implement this strategy, we decompose the computation of $Y = X\mathbf{W}^T$ into two components: the sparse $Y_s$ and low-rank $Y_r$.

**Sparsifying Input Activation**: Firstly, we estimate the threshold for identifying the sparse components of the input $X$. Given a pre-defined sparsity budget $s$, the threshold $t(s)$ is estimated as the $s$th percentile of $X$, *i.e.*, $\mathbb{P}(|X| < t(s)) = s$. Next, we apply the threshold to mask out the low-magnitude channels. The corresponding sparsification function $\sigma_{t(s)}(\cdot)$, is defined as:

$$\sigma_{t(s)}(X)_j := \begin{cases} X_j & \text{if } |X_j| \geq t(s) \\ 0 & \text{if } |X_j| < t(s) \end{cases}$$

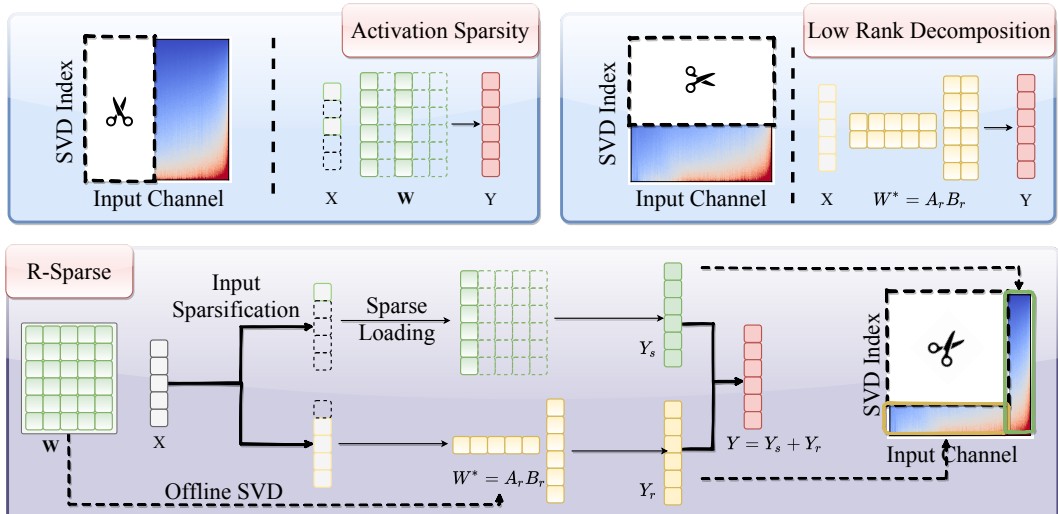

Figure 4: Illustration of various compression techniques with corresponding impact on different input channels and singular values. The horizontal axis of the heatmap represents the input channels, while the vertical axis corresponds to the singular value index.

Note that CATS Lee et al. (2024) employs a similar thresholding strategy to identify sparse components. However, while their approach targets sparsity in the output activation of the gate projection, our method focuses on input sparsity, which can be applied across all linear layers of LLMs.

**R-Sparse Inference**: The original linear layer $Y = X\mathbf{W}^T$ can then be approximated as $Y = Y_s + Y_r$ where $Y_s = \sigma_{t(s)}(X)\mathbf{W}^T$ and $Y_r = (X - \sigma_{t(s)}(X))(\mathbf{A_r B_r})^T$. For the sparse part, we omit unnecessary columns corresponding to input channels with zero values. Additionally, the weights should be stored in a column-major format to enhance memory bandwidth utilization, as GPUs fetch consecutive memory entries during each access. For the low-rank part, we perform SVD on the pretrained weight matrix $\mathbf{W}$ and use its low-rank approximation, where $\mathbf{A_r} = \mathbf{U}_r \Sigma_r^{\frac{1}{2}}$ and $\mathbf{B_r} = \Sigma_r^{\frac{1}{2}} \mathbf{V}^T$, with $r$ representing the selected rank. And we select the most important $r$ components based on the estimated scores in Figure 3. Since this low-rank approximation can be computed offline through a single SVD operation, it won't impact the latency during the inference. The memory I/O overhead is determined by two hyperparameters, $(r, s)$, and is equal to $r\frac{m+n}{mn} + s$ relative to that of a full linear layer. Additionally, we apply R-Sparse inference to all linear layers in both the attention and MLP blocks, aiming to achieve higher sparsity ratios.

### 3.5 OPTIMAL RECIPE FOR SPARSIFICATION

As illustrated in Figure 3, different layers demonstrate varying characteristics of rank-aware sparsity. To more accurately approximate the full computation, we develop an evolutionary strategy to search for the optimal ratio between the sparse and low-rank components within each layer. We begin by defining $\rho_i$, which represents the relative ratio of the sparse part in layer $i$. Given $C_i$ as the sparse budget of layer $i$, the sparse part equals to $s_i = \rho_i C_i$ and the rank is $r_i = (1-\rho_i)C_i \frac{mn}{m+n}$. We employ the search algorithm (Algorithm 1) to obtain the optimal $\rho^* = \{\rho_1^*, \rho_2^*, \ldots, \rho_L^*\} = \arg\min_\rho \mathcal{L}(f, \rho)$, where the loss $\mathcal{L}$ is the average perplexity over 16 randomly selected samples from the C4 training set and $f$ is the original LLMs. We retain the individuals with lower perplexity at each generation. To expedite the convergence of the search process, we implement a group-wise strategy with a group size of 28. In this approach, we optimize the variables of one group at a time, while holding the variables of the other groups at the values from the most recent best-performing individual.

The population size is set to 32, with both the mutation rate $p_m$ and crossover rate $p_c$ equals 0.5, and the total number of generations is 5. The overhead of the search process is minimal, taking approximately one hour on a single A6000 GPU for the Llama-2-7B model.

---

**Algorithm 1** Search Algorithm for Sparsification Recipe

---

1: **Initialize:** A pre-trained LLM $\mathcal{M}$ that consists of $L$ layers. A population size of $\mathcal{P}$, mutation rate $p_m$, crossover rate $p_c$, a total of $\mathcal{T}$ generations.
2: Randomly initialize population $\mathcal{G} = \{\rho^1, \rho^2, ..., \rho^{\mathcal{P}}\}$ where $\rho^i = \{\rho_1^i, \rho_1^i, ..., \rho_L^i\}$
3: $S = \text{Best}(\mathcal{G}); \hat{\mathcal{G}} = \{\}$          ▷ Select the best individual from the group
4: **for** generation $t = 1, ..., \mathcal{T}$ **do**
5:      **for** generation $i = 1, ..., \mathcal{P}$ **do**
6:          $m^i = \rho^{x_1} + p_m(\rho^{x_2} - \rho^{x_3})$    ▷ Mutation: $x_1, x_2, x_3$ are randomly chosen from $\{1, 2, ..., \mathcal{P}\}$.
7:          $\hat{\rho}^i = (\alpha > p_c)m^i + (\alpha \leq p_c)\rho^i$        ▷ Crossover: $\alpha$ is random variables from $(0, 1)^L$
8:          $\hat{\mathcal{G}} = \hat{\mathcal{G}} \cup \{\hat{\rho}^i\}$
9:      **end for**
10:      $\mathcal{G} = \text{Top\_K}(\hat{\mathcal{G}} \cup \mathcal{G}); S = \text{Best}(\mathcal{G}); \hat{\mathcal{G}} = \{\}$        ▷ Select the next generation
11: **end for**
12: **Return:** Best recipe $S$.

---

# 4 EXPERIMENTS

Table 1: Comparison between `R-Sparse` and other baselines on common-sense reasoning tasks.

| Models | WG | PIQA | SciQ | OBQA | HS | BoolQ | Arc-E | Arc-C | Average |
|---|---|---|---|---|---|---|---|---|---|
| Llama-2-7B | 69.14 | 78.07 | 93.80 | 31.40 | 57.13 | 77.71 | 76.35 | 43.43 | 65.88 |
| `ReLUfication` | 49.25 | 54.19 | 25.90 | 15.40 | 25.82 | 60.00 | 27.90 | 24.23 | 35.34 |
| `CATS`$_{22\%}$ | 67.72 | 77.37 | 92.80 | 30.40 | 57.03 | 72.87 | 74.71 | 41.64 | 64.32 |
| `CATS`$_{40\%}$ | 55.01 | 66.97 | 57.20 | 20.20 | 36.27 | 62.81 | 44.02 | 27.56 | 46.26 |
| `R-Sparse`$_{40\%}$ | 68.03 | 77.31 | 93.90 | 30.80 | 55.62 | 75.99 | 75.67 | 42.66 | 65.00 |
| `GRIFFIN`$_{33\%}$ | 62.04 | 71.27 | 89.00 | 22.00 | 47.20 | 60.98 | 60.94 | 32.00 | 55.68 |
| `GRIFFIN`$_{50\%}$ | 53.59 | 64.74 | 77.70 | 17.40 | 35.64 | 56.42 | 40.74 | 21.08 | 45.91 |
| `R-Sparse`$_{50\%}$ | 67.40 | 77.31 | 93.90 | 31.40 | 54.26 | 72.84 | 74.58 | 40.78 | 64.06 |
| Llama-3-8B | 72.69 | 79.71 | 96.20 | 34.80 | 60.18 | 81.35 | 80.09 | 50.51 | 69.44 |
| `ReLUfication` | 50.83 | 53.48 | 22.20 | 14.80 | 25.53 | 52.78 | 24.45 | 21.50 | 33.20 |
| `CATS`$_{22\%}$ | 70.17 | 79.00 | 94.90 | 31.20 | 57.81 | 76.51 | 75.29 | 46.50 | 66.42 |
| `CATS`$_{40\%}$ | 48.22 | 56.96 | 36.90 | 16.20 | 27.30 | 49.05 | 30.30 | 22.35 | 35.91 |
| `R-Sparse`$_{40\%}$ | 71.11 | 78.24 | 95.90 | 34.60 | 58.33 | 79.85 | 79.67 | 49.23 | 68.37 |
| `GRIFFIN`$_{33\%}$ | 63.54 | 71.87 | 89.40 | 24.00 | 48.08 | 54.34 | 62.33 | 34.73 | 56.04 |
| `GRIFFIN`$_{50\%}$ | 52.80 | 64.74 | 73.90 | 19.20 | 35.62 | 47.61 | 43.64 | 23.38 | 45.11 |
| `R-Sparse`$_{50\%}$ | 69.30 | 77.69 | 96.00 | 31.60 | 56.64 | 76.73 | 76.94 | 44.71 | 66.20 |
| Mistral-7B | 74.11 | 80.41 | 96.00 | 32.20 | 61.05 | 83.85 | 80.68 | 50.85 | 69.89 |
| `ReLUfication` | 48.62 | 51.52 | 23.51 | 14.40 | 25.81 | 42.02 | 27.82 | 24.06 | 32.22 |
| `CATS`$_{22\%}$ | 72.22 | 79.82 | 94.30 | 32.20 | 60.79 | 80.46 | 77.78 | 50.51 | 68.51 |
| `CATS`$_{40\%}$ | 50.83 | 58.05 | 28.60 | 19.20 | 28.06 | 60.21 | 30.22 | 25.43 | 37.58 |
| `R-Sparse`$_{40\%}$ | 72.45 | 79.49 | 96.10 | 30.40 | 59.68 | 82.11 | 79.80 | 47.18 | 68.40 |
| `GRIFFIN`$_{33\%}$ | 63.30 | 75.95 | 91.00 | 25.40 | 53.01 | 64.28 | 68.73 | 36.60 | 59.78 |
| `GRIFFIN`$_{50\%}$ | 54.22 | 67.90 | 79.10 | 19.80 | 39.99 | 47.31 | 49.83 | 26.37 | 48.07 |
| `R-Sparse`$_{50\%}$ | 72.69 | 79.92 | 96.10 | 30.60 | 58.94 | 82.81 | 78.91 | 47.18 | 68.39 |

## 4.1 GENERAL SETUP

**Models and Datasets.** To evaluate the effectiveness of `R-Sparse`, we consider three representative large language model (LLM) families: Llama-2 (Touvron et al., 2023), Llama-3 (Dubey et al., 2024), and Mistral (Jiang et al., 2023). We assess the models on several popular tasks, including eight common-sense reasoning tasks: Winogrande (WG) (Sakaguchi et al., 2021), PIQA (Bisk et al., 2020), SciQ (Welbl et al., 2017), OpenBookQA (OBQA) (Mihaylov et al., 2018b), HellaSwag (HS) (Zellers et al., 2019), BoolQ (Clark et al., 2019), and ARC (ARC-Easy and ARC-Challenge) (Clark et al.,

2018b). Evaluations are conducted using the lm-evaluation-harness framework (Gao et al., 2021). Additionally, we report results on text summarization tasks using XSUM (Narayan et al., 2018), as well as language modeling tasks on the validation set of WikiText-2 (Merity et al., 2016). For common-sense reasoning, we report accuracy, while summarization tasks are evaluated via Rouge-L scores and language modeling is assessed by perplexity.

**Baselines.** Since `R-Sparse` does not require additional training, we compare it against several competitive training-free methods. (i) `ReLUfiction` (Mirzadeh et al., 2023) where the non-ReLU activation functions in the MLP block are replaced with ReLU, and accuracy is reported without retraining. (ii) `CATS`(Lee et al., 2024) that sparsifies $\mathbf{W}_{up}$ and $\mathbf{W}_{down}$ based on the magnitude of output activations from $\mathbf{W}_{gate}$. (iii) `GRIFFIN` (Dong et al., 2024): It sparsifies all layers in the MLP block, selecting important channels based on statistics from the pre-filling stage. Different from `CATS` and `GRIFFIN`, which focus only on the MLP blocks, `R-Sparse` sparsifies all linear layers, including the attention blocks. For a fair comparison, we report performance with the original reported sparsity ratios (50% for the sparsified modules, corresponding to 22% model-level sparsity for `CATS` and 33% for `GRIFFIN`). We also compare the results with higher sparsity ratio by scaling up the MLP block sparsity for both methods. All sparsity ratios reported in the following experiments are measured at the model level. More details are included in Appendix A and B.1.

## 4.2 END-TO-END RESULTS

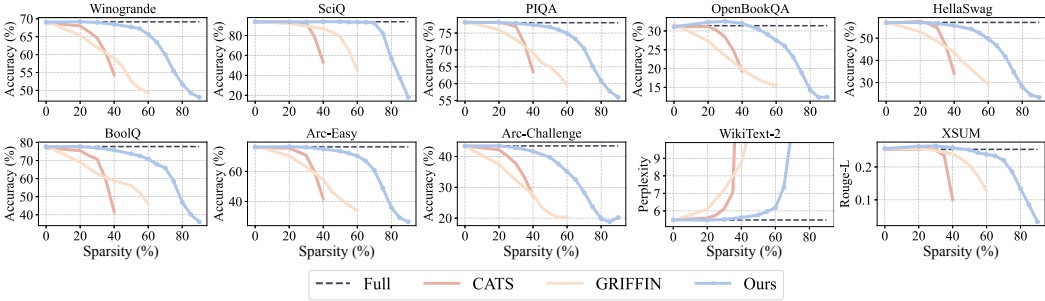

Figure 5: Comparison results of Llama-2-7B across different model-level sparsity ratios on common-sense reasoning, language modeling and summarization tasks.

We begin by presenting the end-to-end performance of `R-Sparse` and baseline methods across different models, tasks, and sparsity ratios. The results, shown in Table 1 and Figure 5, highlight several key observations: (I) `R-Sparse` consistently outperforms `CATS` (Lee et al., 2024) and `GRIFFIN` (Dong et al., 2024) across all common-sense reasoning, language modeling, and summarization tasks. With the same model-level sparsity budget (*i.e.* `CATS`$_{40\%}$ *v.s.* `R-Sparse`$_{40\%}$ and `GRIFFIN`$_{50\%}$ *v.s.* `R-Sparse`$_{50\%}$), `R-Sparse` achieves an average performance gain of $18.74\%$ over `CATS` and $18.15\%$ over `GRIFFIN` on Llama-2-7B. This improvement primarily stems from three factors: ❶ while `CATS` and `GRIFFIN` only sparsify the MLP block, `R-Sparse` can be applied to both the attention and MLP blocks; ❷ we extends standard activation sparsity with rank-aware sparsity, providing a better approximation of the full computation; ❸ and we further leverages the adaptive rank properties of different layers by searching the optimal sparse-rank ratio $\rho$. Detailed ablation studies on these factors are discussed in Section 4.4. (ii) `R-Sparse` achieves performance comparable to the full model with minimal degradation at a sparsity ratio around 50% while in some tasks, *e.g.*, SciQ, a matching performance can be achieved even at a sparsity ratio of 70%. (iii) For some tasks, a moderate sparse treatment slightly enhances the accuracy, such as $1.60\%$ improvements at 30% sparsity ratio on the OpenBookQA task.

## 4.3 EFFICIENCY

We demonstrate the end-to-end efficiency improvements of `R-Sparse`. For this, we collected five samples that consists of 2048 tokens and generate new content ranging in length from 128 to 2048 tokens to evaluate performance across different generation lengths. Without losing generality,

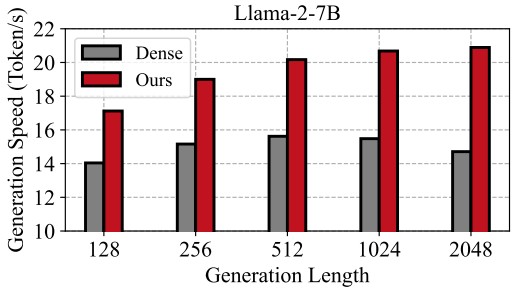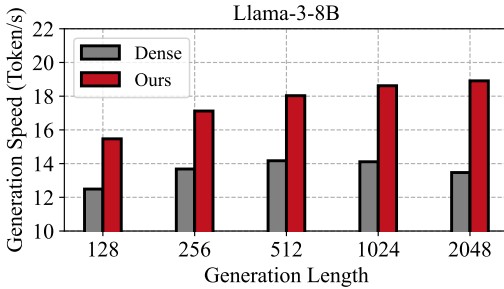

Figure 6: Generation speeds of Llama-2-7B and Llama-3-8B using a uniform 50% sparsity in our method. The prompts consist of 2048 tokens, with generation lengths ranging from 128 to 2048. The generation speed is calculated as the number of generated tokens divided by the total generation time.

Table 2: Compatibility with weight quantization.

| Methods | WG | PIQA | SciQ | OBQA | Average |
|---|---|---|---|---|---|
| FP16 | 69.14 | 78.07 | 93.80 | 31.40 | 68.10 |
| INT4 | 68.19 | 77.48 | 93.80 | 29.80 | 67.32 |
| R-Sparse$_{40\%}$ | 68.03 | 77.31 | 93.90 | 30.80 | 67.51 |
| R-Sparse$_{50\%}$ | 67.40 | 77.31 | 93.90 | 31.40 | 67.50 |
| INT4 R-Sparse$_{40\%}$ | 66.93 | 76.71 | 92.80 | 29.20 | 66.41 |
| INT4 R-Sparse$_{50\%}$ | 66.38 | 75.95 | 92.10 | 28.60 | 65.76 |

Table 3: Results of sparse and low-rank baselines.

| Methods | WG | PIQA | SciQ | OBQA | Average |
|---|---|---|---|---|---|
| Full | 69.14 | 78.07 | 93.80 | 31.40 | 68.10 |
| Sparse | 65.11 | 77.37 | 93.30 | 29.20 | 66.25 |
| Low-Rank | 49.88 | 53.32 | 14.80 | 14.20 | 33.05 |
| R-Sparse | 67.40 | 77.31 | 93.90 | 31.40 | 67.50 |

our implementation is based on the Hugging Face library with FP32 precision data format. All experiments are conducted on a single NVIDIA A6000 GPU without offloading. We applied a uniform 50% sparsity to R-Sparse, achieving comparable performance as shown in Section 4.2 and utilized a customized Triton kernel to reduce data transfer between on-chip SRAM and HBM. As illustrated in Figure 6, R-Sparse achieved up to 42% and 40% improvements in generation speed for Llama-2-7B and Llama-3-8B, respectively, highlighting the effectiveness of our approach.

## 4.4 ABLATION STUDY AND FURTHER INVESTIGATION

We conduct extensive ablation studies of R-Sparse, summarized by the following research questions: *Q1*: Is R-Sparse compatible with weight quantization? *Q2*: How does R-Sparse compare with vanilla activation sparsity? *Q3*: What's the benefit of optimal sparsification recipe?

*A1*: **Compatible with quantization.** We demonstrate that R-Sparse is highly compatible with weight quantization. As shown in Table 2, when combined with 4-bit quantization, R-Sparse achieves an average accuracy of 66.41% at 40% sparsity and 65.76% at 50% sparsity on common-sense reasoning tasks, closely comparable to the full model's performance of 68.10% and the quantization-only result of 67.32%. Note that we use GPTQ (Frantar et al., 2022) for weight quantization with a group size of 128, that provides matching performance as the full baseline. The compatibility of R-Sparse with weight quantization offers further potential efficiency gains through optimized CUDA kernels that fuse the sparse and quantization operations.

*A2*: **R-Sparse outperforms both vanilla activation sparsity and low-rank decomposition.** Table 3 compares R-Sparse with vanilla activation sparsity (Sparse) and low-rank decomposition (Low-Rank). For the sparse and low-rank baselines, we apply the sparsification operation on all linear layers, maintaining the same model-level sparsity ratios for each method. Experiments conducted with 50% sparsity on Llama-2-7B show that R-Sparse consistently outperforms the Sparse baseline, with an average improvement of 0.98%, while the Low Rank method fails to maintain performance. This is expected, as the low-rank properties vary across layers: layers with intrinsic low-rank characteristics can be well-approximated with a small $\rho$, while higher-rank layers benefit from higher sparse components, leading to a higher $\rho$. With that, R-Sparse combines both scenarios and provides a more effective approximation.

***A3*: Further enhancement through better sparsification recipes.** We compare the searched sparsification recipes with uniform ones. For the uniform approach, we set $\rho = 0.95$ uniformly across all layers, based on a grid search using 16 training samples from the C4 dataset. In contrast, the adaptive strategy is based on the search algorithm. As shown in Table 4, the evolutionary search algorithm outperforms the uniform strategy, achieving up to a 1.60% accuracy gain across sparsity ratios ranging from 40% to 70%. Notably, at

Table 4: Comparison of different sparsification recipes.

| Tasks | Methods | 40% | 50% | 60% | 70% | Average |
|-------|---------|-----|-----|-----|-----|---------|
| OBQA | Uniform | 29.80 | 30.40 | 24.60 | 21.40 | 26.55 |
| | Adaptive | 30.80 | 31.40 | 27.80 | 24.00 | 28.50 (+1.95) |
| Arc-E | Uniform | 74.92 | 74.03 | 68.69 | 60.77 | 69.60 |
| | Adaptive | 75.67 | 74.58 | 70.29 | 61.41 | 70.49 (+0.89) |
| Arc-C | Uniform | 41.30 | 39.08 | 35.49 | 28.16 | 36.01 |
| | Adaptive | 42.66 | 40.78 | 36.01 | 29.10 | 37.14 (+1.13) |
| BoolQ | Uniform | 75.32 | 72.54 | 69.42 | 63.85 | 70.28 |
| | Adaptive | 75.99 | 72.84 | 72.14 | 64.59 | 71.39 (+1.11) |

higher sparsity ratios, the adaptive strategy yields greater performance improvements. For example, on the OpenBookQA task, at the 70% sparsity ratio, there is a 2.60% gain compared to a 0.80% improvement at the 50% sparsity ratio.

## 5 CONCLUSION

In this paper, we focus on the activation sparsity of the input side. By leveraging the intrinsic sparse structure within input activations and singular value components, we introduce `R-Sparse`, which eliminates the need for extensive pre-training and predicting active output channels, achieving 50% model-level sparsity without additional training. Experiments across different LLM families, including Llama-2, Llama-3, and Mistral, demonstrate the effectiveness of `R-Sparse`—achieving comparable performance at 50% sparsity across ten common-sense reasoning, language modeling, and text summarization tasks. This high sparsity ratio also brings a significant 43% speed improvement with a customized kernel. Our work demonstrates that high levels of sparsity can be achieved in both the attention and MLP blocks of advanced LLMs without any performance loss, benefiting the further deployment of LLMs on edge devices.

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

## A MORE IMPLEMENTATION DETAILS

In the experiments, the sparsification techniques are applied exclusively during the decoding stage. For the tasks involving only a single-step decoding phase, original GRIFFIN implementation only apply the sparsification on the final token while in our experiments, we simulate the first half of the prompt as the prefilling stage, applying sparsification to the second half to more effectively evaluate the generation capabilities of LLMs.

## B EXTENDED EXPERIMENTS

### B.1 SCALING UP SPARSITY RATIOS OF GRIFFIN

For GRIFFIN, we explore two strategies for scaling up the model-level sparsity ratios: (i) MLP, where we directly increase the sparsity ratios within the MLP blocks and report the resulting model-level sparsity; and (ii) All, where we extend the strategy to include attention blocks. In this case, we use the same metrics to identify important channels based on the activations during the prefilling stage and determine the corresponding active channels during the decoding stage. Results are presented in Figure 7 where the MLP strategy is significantly better than the All. Thus in the main context, we report the results of MLP strategy for GRIFFIN.

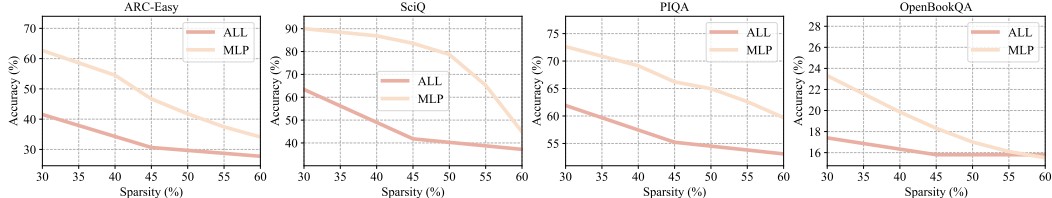

Figure 7: Results of GRIFFIN with Llama-2-7B.

### B.2 RANK-AWARE ACTIVATION SPARSITY ACROSS VARIOUS DATASETS AND DIFFERENT NUMBER OF SAMPLES

We extend the observations from Figure 3 to additional datasets and varying numbers of samples. The results are presented in Figure 8 and Figure 9. Across different numbers of training samples, the importance patterns consistently exhibit high sparsity. Additionally, to ensure data diversity, we evaluated different domains from the RedPajama dataset[1], including GitHub, ArXiv, StackExchange, and Wikipedia. As shown in Figure 9, the importance patterns are remarkably similar across these datasets, demonstrating the generalization capability of the R-Sparse approach.

---

[1]The training data is obtained from `https://huggingface.co/datasets/togethercomputer/RedPajama-Data-1T`

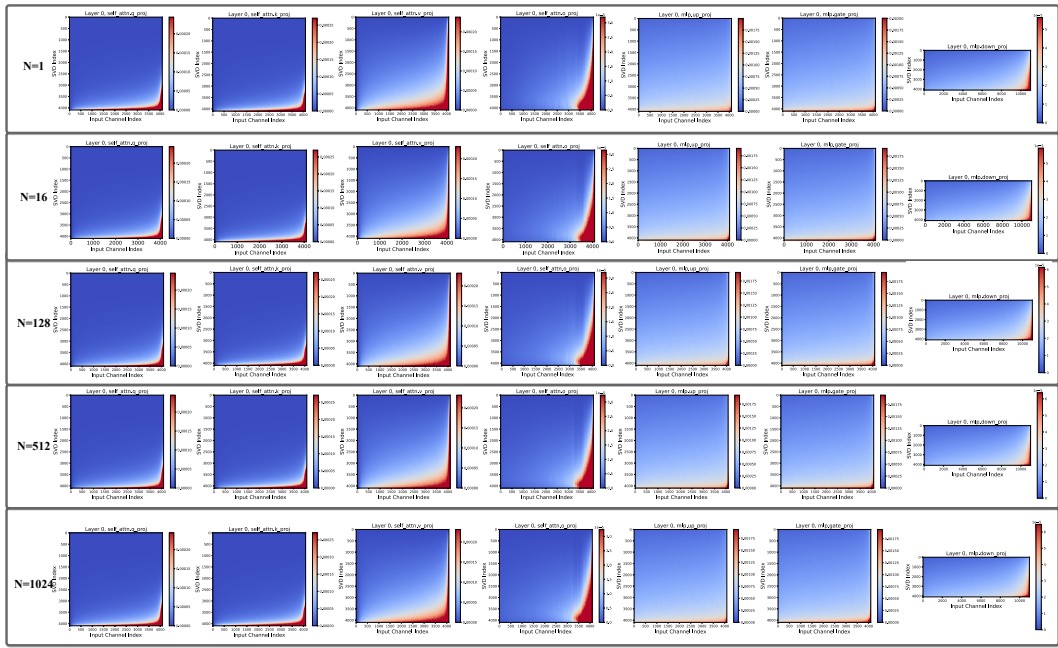

Figure 8: Importance of each input channel and singular value across varying samples. The number of samples ranging from 1 to 1024. Results are collected from Llama-2-7B model from C4 training set. The sequence length of each sample equals to 4096.

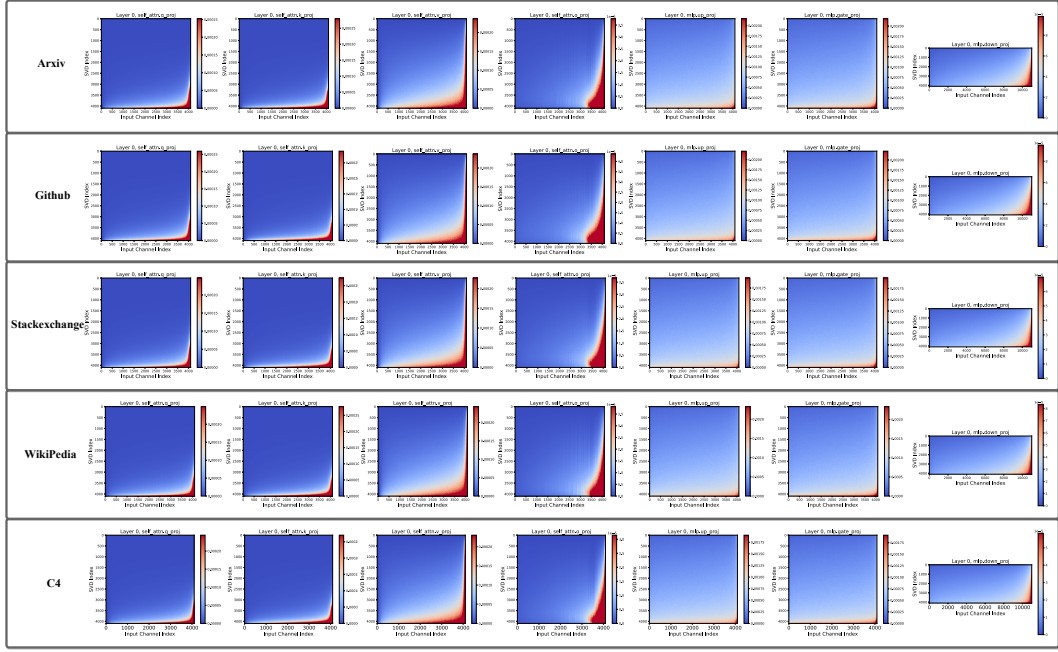

Figure 9: Importance of each input channel and singular value components across different datasets.

