# OpenReview forum: "R-Sparse: Rank-Aware Activation Sparsity for Efficient LLM Inference"
_ICLR.cc/2025/Conference — ICLR 2025 Poster_

### Official Review · Reviewer_sWLw · 2024-11-02

**Soundness:** 3
**Presentation:** 3
**Contribution:** 2
**Rating:** 6
**Confidence:** 4

**Summary:**

This paper proposes a rank-aware activation sparsity, including applying input sparsification and weight decomposition. Experiments show that the proposed R-Sparse improves end-to-end efficiency while maintaining comparable sparsity.

**Strengths:**

1. The proposed methods are easy to understand.
2. The experimental results are good.

**Weaknesses:**

1. The connections between motivations and the proposed methods are weak. For example, the first motivation claims that the outputs for the non-sparse inputs can be regarded as biases. However, it is not sure which part of the proposed methods is motivated by this. Please explicitly explain how the observation about non-sparse inputs being treated as biases directly informs specific components of their R-Sparse method.

2. The contributions are incremental, which only directly apply existing techniques, such as CATS for sparsity, SVD for weight decomposition, and genetic algorithm for hyperparameters searching. Please more clearly articulate the novelty of your approach. Is there any specific design in SVD?

3. The paper claims that existing non-ReLU activations such as SiLU and GELU introduce less sparsity. However, the experiments lack generality as Llama 2/3 and Mistral all adopt SiLU as activation functions. For example, adding results of models using GELU activation functions such as Gemma would be helpful.

**Questions:**

See weakness

---

> ### Author Response · Authors · 2024-11-22
> **Responses to Reviewer sWLw**
>
> Many thanks to Reviewer sWLw for the constructive suggestions, which have helped us further improve the quality of our work. Below, we provide detailed responses to address the concerns.
>
> **[Q1: Connection between the motivation and the method]** Thanks for the question. We’d like to clarify that the motivation for Case I (Non-sparse components as biases) primarily serves as an initial investigation for Case II (Rank-aware activation sparsity). The observation of rank-aware activation sparsity then directly supports the R-Sparse method.
>
> As discussed in Line 200, obtaining the non-sparse bias terms is computationally expensive, while each individual bias term only contributes a rank-1 component. This motivated us to investigate the overall rank across multiple bias terms from different inputs. We discovered that the space spanned by these biases across thousands of tokens exhibits a low-rank structure, highlighting the relationship between low-rank decomposition and activation sparsity.
>
> Further, we examined the importance patterns of each input channel and singular value, as illustrated in Figure 3. These rank-aware sparsity patterns validate and underpin the core methodology of our framework, demonstrating its efficiency and effectiveness.
>
>
> **[Q2: Clarifying contribution]** We respectfully disagree that our contribution is incremental. First, as discussed in Lines 51–64, one significant challenge in prior activation sparsity methods is the difficulty of predicting active channels. For instance, CATS utilizes the output of mlp.gate to identify active channels for sparsifying mlp.up and mlp.down, while GRIFFIN determines active channels based on output features from the pre-filling stage. In contrast, our method observes and exploits the inherent low-rank and sparse structures within the input features. This enables us to sparsify all linear layers without relying on prediction mechanisms, overcoming the limitations of prior methods. Specifically, CATS is restricted to sparsifying mlp.gate and mlp.down, and GRIFFIN applies sparsity only to MLP blocks. Our approach provides a broader and more effective solution to activation sparsity, making a substantial advancement over existing methods.
>
> Additionally, the inherent low-rank and sparse structures within the input features contribute to significantly enhancing the achievable sparsity levels for non-ReLU-based LLMs. In comparison, methods like CATS and GRIFFIN are limited to achieving model-level sparsity ratios of only 22% to 33%, whereas our R-Sparse method achieves approximately 50% sparsity. Thus, we believe our R-Sparse makes a significant contribution to alleviate the main challenges of previous activation sparsity approaches, that is: (i) feasibility for non-ReLU based LLMs; (ii) difficulty in predicting active channels; and (iii) limited sparsity levels.
>
>
> **[Q3: Experiments with GELU activation]** Great suggestion! We conducted additional experiments to evaluate our method on the Gemma-7B model. The results, presented in Table R1, show that our method significantly outperforms both CATS and GRIFFIN, achieving an average accuracy improvement of 21.76% and 4.62% at 40% and 50% sparsity, respectively.
>
> Table R1 Comparison between R-Sparse and other baselines on common-sense reasoning tasks.
>
> | Gemma-7B          | WG     | PIQA   | SCIQ   | OBQA   | HS     | BOOLQ  | ARC-E  | ARC-C  | AVG    |
> |----------------|--------|--------|--------|--------|--------|--------|--------|--------|--------|
> | CATS 40%       | 58.01  | 66.54  | 51.9   | 19.8   | 37.64  | 62.11  | 42.97  | 27.9   | 45.86  |
> | R-Sparse 40%   | 72.22  | 78.89  | 94.3   | 33.2   | 59.68  | 74.04  | 80.39  | 48.21  | 67.62  |
> | GRIFFIN 50%    | 67.56  | 74.81  | 77.1   | 25.2   | 54.14  | 62.29  | 67.97  | 40.44  | 58.69  |
> | R-Sparse 50%   | 69.46  | 78.35  | 87.6   | 29.6   | 57.3   | 63.03  | 75.81  | 45.31  | 63.31  |

---

> > ### Comment · Reviewer_sWLw · 2024-11-25
> >
> > Thanks for your responses, which address my concerns. I will raise my score.

---

> > > ### Author Response · Authors · 2024-11-25
> > > **Thanks for the response**
> > >
> > > We sincerely appreciate all your constructive feedback and positive evaluations. Thanks for your time and support!

---

### Official Review · Reviewer_LhWB · 2024-11-04

**Soundness:** 3
**Presentation:** 2
**Contribution:** 3
**Rating:** 5
**Confidence:** 4

**Summary:**

The authors propose a novel training-free activation sparsity method called R-sparse. This method is applicable to non-ReLU-based large language models (LLMs) and eliminates the need for prediction by utilizing input activation sparsity. Furthermore, it is a method that can be applied not only to MLP modules but also to attention modules.

**Strengths:**

- The authors successfully identified existing challenges and, in connection to these, suggested a training-free, prediction-free activation sparsity method. Furthermore, their method is applicable to attention modules.
- Through various experiments, they demonstrated performance improvements.

**Weaknesses:**

- difficult to read (clearity)
  - In introduction, how should Figure 1 be interpreted? It is difficult to understand the meaning of the figure until reviewing the explanation in Section 3.3.
  - In Section 3.2, what does the term "bias" mean? Is it interpreted as "bias" in the sense that performance is maintained even after replacing non-sparse values smaller than 0 with a constant? Does the term also include non-sparse components greater than 0? In Lines 183-185, why is it defined as "sparse", when $H_k \geq T_0$?

- the appropriateness of the title.
  - The proposed method appears to be "activation sparsity, then low-rank decomposition for non-important activations" rather than using both approaches simultaneously (rank-aware activation sparsity).



==== After Rebuttal ====
- I understand. However, the terms in Section 3.2 should be clarified more explicitly.
- It might be beneficial to add an explanation using examples, when $T_0 = 0$.
  - positive = sparse components = not pruned values
  - negative = non-sparse components = to be pruned values
- In fact, the most confusing term is "sparse components." It would be beneficial to clearly indicate, as suggested in the response, that it originates from prior research.

**Questions:**

- (Writing) Please switch the positions between mlp.up_proj and mlp.gate_proj of layer 0 in Figure 3.
- Recommended reference
  - Alizadeh, Keivan, et al. "Llm in a flash: Efficient large language model inference with limited memory."
- Definition of multi-phase ReLU.
  - Is the multi-phase ReLU expressed on line 170 correctly defined? Where is sparsity defined? Shouldn't there be a definition for when x $< T_{L}$? Moreover, the output should get closer to zero as it becomes more negative, but it is defined in the opposite direction.
  - For instance, $T_0 = 0, T_1=-1, T_2=-2$, please explain it.

---

> ### Author Response · Authors · 2024-11-22
> **Responses to Reviewer LhWB**
>
> We sincerely thank Reviewer LhWB for the detailed and insightful suggestions. To enhance the writing quality and address the concerns raised, we provide point-by-point responses below:
>
> **[Q1: Explanation about Figure 1.]**: Thanks for the suggestion. We have revised the caption of Figure 1 in the updated draft to improve clarity and ensure it is easier to understand. The new caption is: “Contributions of each input channel and singular value components. The measurement metric is detailed in Section 3.3. Results are obtained from Llama-2-7B with 16 training samples from C4. Both the input channel and SVD components are sorted from small to large for better visualization.
>
>
> **[Q2: Explanation about the “Bias” term in Section 3.2.]**: The bias term includes all the non-sparse components where $H_k < T_0$. Its definition depends on the value of $T_0$ rather than being strictly tied to $O$. If $T_0$ is greater than $0$, the non-sparse components include values that are both greater than and less than $O$.
>
> We refer to this term as “bias” because the output $Y=HW_{down}^T$ can be interpreted as a weighted linear combination of the columns in $W_{down}^T$, where the coefficient of column $k$ equals $H_k$. For $H_k \geq T_0$, the calculation cannot be simplified, as each column is weighted by a different coefficient, and these are therefore defined as sparse components.
>
> In contrast, for $H_k < T_0$, the columns satisfying $T_{i+1} \leq H_k < T_i$ share the same coefficient after the multi-phase ReLU activation function. These components can be viewed as a bias term, where the weighting is given by $\frac{T_j + T_{j+1}}{2}$.
>
> **[Q3: Appropriateness of paper title.]**: Thanks for pointing this out. We’d like to clarify that the reason for using rank-aware activation sparsity is because for different linear layers, the activation sparsity is affected by its low-rank properties, thus we need to co-design the sparsity and low-rank decomposition to better preserve original functionality. We achieve this through an evolutionary search approach, which effectively balances these factors for optimal performance. We’re willing to adjust the title if Reviewer LhWB has further suggestions.
>
> **[Q4: Modification of Figure 3.]**: Thanks for the careful reading. The modified Figure 3 is updated in the draft.
>
>
> **[Q5: More reference.]**: Thanks for the suggestion. This work provides an interesting and solid hardware solution for sparsity-aware memory loading, that is orthogonal and provides a useful tool to further implement our methods on edge-devices. We’ve included the reference in the updated PDF.
>
>
> **[Q6: Definition of multi-phase ReLU.]**: Great Catch! In our investigation, we set $T_{l-1}$ as the minimum value of input, to avoid scenarios when $x < T_{l-1}$. And the sparsity is defined as the ratios of $x < T_0$. In this way, for all $x < T_0$, the output will be the nearest discrete value of $\frac{T_i + T_{i+1}}{2}$, to better preserve the original functionality. We have clarified this explanation in the updated draft to avoid misleading.

---

> > ### Author Response · Authors · 2024-11-25
> > **We are keen to discuss further with you**
> >
> > Dear Reviewer LhWB,
> >
> > Thank you for taking the time to review our work. Your suggestions have been incredibly helpful in improving the writing quality and ensuring our work is more easily understood. We have carefully addressed each of the writing issues to avoid misunderstandings. The revisions have been updated in the PDF, with the modified content marked in red.
> >
> > Could you please review the responses and let us know if there are any additional questions or concerns? Thanks again for your valuable feedback.
> >
> > Best,
> >
> > Authors

---

> > > ### Author Response · Authors · 2024-11-26
> > > **We are keen to discuss further with you**
> > >
> > > Dear Reviewer LhWB,
> > >
> > > Thank you again for your careful reading and thoughtful feedback on our work. As your comments primarily focused on improving the writing, we have made every effort to address them thoroughly.
> > >
> > > At your convenience, could you please review our responses and let us know if you have any additional questions or concerns? Thank you!
> > >
> > > Sincerely,
> > >
> > > The Authors

---

> > > > ### Author Response · Authors · 2024-11-28
> > > >
> > > > Dear Reviewer LhWB,
> > > >
> > > > We sincerely appreciate the time you have taken to review our work. We have carefully addressed all your comments, which primarily focused on writing improvements. Could you kindly review the updates and let us know if you have any further questions? Thank you!
> > > >
> > > > Sincerely,
> > > >
> > > > Authors

---

> > > > > ### Author Response · Authors · 2024-11-29
> > > > > **We are keen to discuss further with you**
> > > > >
> > > > > Dear Reviewer LhWB,
> > > > >
> > > > > Thank you again for your time and efforts in reviewing our work. As the discussion period deadline is approaching, could you kindly review our responses and let us know if you have any further questions?
> > > > >
> > > > > Thank you!
> > > > >
> > > > > Sincerely,
> > > > >
> > > > > The Authors

---

> ### Comment · Reviewer_LhWB · 2024-12-01
>
> Thank you for your kind response. There is one unclear part, and I would like to ask a question about it.
>
> **[Q2 and Q6]**
> - The authors said that "the sparsity is defined as the ratios of $x < T_{0}$", in Q6.
>   - However, within this range, $x$ actually takes on the nearest discrete value rather than 0 in order to maintain the original functionality. Is it correct to define this as "sparsity"?
>   - As requested in the original question, please provide an example to explain the situation. I understand the case where the minimum of $x$'s input is -2.
> - Futhermore, the authors said that "all the **non-sparse** components where $H_{k} < T_{0}$", in Q2.
>   - Is not this expression contradictory to the definition provided above?
>
> Please clarify it. Thanks.

---

> ### Author Response · Authors · 2024-12-02
> **Thanks for the responses**
>
> Thank you for the detailed feedback. The sparsity definition we use aligns with the spirit of previous network pruning works [1-3] where the sparse sub-network refers to the remaining sub-network, and the corresponding parameter values are unchanged after the pruning operation. The sparsity ratio is defined as the proportion of parameters that their values are changed after pruning (set to zero). Intuitively, the sparse components represent the critical parts of the network, while the non-sparse components correspond to the non-critical parts.
>
> In our definition, the non-sparse components are those with $H_k < T_0$, as these typically have a lesser impact on the original functionality, making them the non-critical part. Conversely, the sparsity ratio measures the proportion of elements that are altered after applying the multi-phase ReLU (i.e., $x<T_0$). Thus, the two concepts are not contradictory.
>
> Regarding the question, “x actually takes on the nearest discrete value rather than 0 in order to maintain the original functionality. Is it correct to define this as ‘sparsity’?” — Yes, this definition differs from the strict interpretation of sparsity, where non-critical values are explicitly set to zero. However, it aligns with a broader concept of sparsity. Here, sparsity refers to the effective reduction of non-critical elements, even if these values are adjusted to discrete values rather than being completely zeroed out, to preserve functionality.
>
> For further clarification, we provide a conceptual example in Table R1. The table shows an input feature of size 10, with the values of each element reported before and after applying the multi-phase activation function.
>
> Table R1, examples of values before and after multi-phase activation function. Where $T_0=0$, $T_1=-1$ and $T_2=-2$ and the minimum values of input features is -2. $s$ represents elements that belong to the sparse components, while (non-$s$) denotes the non-sparse components.
>
> | Category | s | s | non-s | non-s | s | non-s | non-s   | non-s | s | s |
> | ------ | ---- | ---- | ----- | ----- | ---- | ----- | ---- | ----- | ---- | ---- |
> | Values Before | 1.73 | 3.55 | -0.43 | -0.27 | 7.59 | -1.05 | -2   | -1.77 | 9.43 | 6.82 |
> | Values After  | 1.73 | 3.55 | -0.5  | -0.5  | 7.59 | -1.5  | -1.5 | -1.5  | 9.43 | 6.82 |
>
> [1] The Lottery Ticket Hypothesis: Finding Sparse, Trainable Neural Networks.
>
> [2] Rethinking the Value of Network Pruning.
>
> [3] What is the State of Neural Network Pruning?

---

> > ### Author Response · Authors · 2024-12-02
> >
> > Dear Reviewer LhWB,
> >
> > Thank you again for your careful review. With the discussion period ending in less than 24 hours, could you kindly review our responses and let us know if you have any further questions? Thank you!
> >
> > Sincerely,
> >
> > The Authors

---

### Official Review · Reviewer_zFFb · 2024-11-04

**Soundness:** 3
**Presentation:** 3
**Contribution:** 3
**Rating:** 6
**Confidence:** 2

**Summary:**

This paper uses the inherent sparsity and low-rank properties of input activations in LLMs to accelerate the inference of LLMs. It sparsifies the input activations to remove unnecessary activations, and interprets unnecessary activations as a bias term, using SVD to compensate for this bias. The proposed method achieves higher sparsity than the baseline while maintaining overall performance.

**Strengths:**

- This paper proposes a training-free activation sparsification method to speed up LLM inference. The proposed method can maintain performance at the same level as full parameters at sparsity of about 50%.
- This paper proposes to design different sparsity ratios for different layers to improve overall performance.
- This paper is very clearly written and the proposed method is easy to follow.

**Weaknesses:**

- The analysis of "the contribution of each input channel and singular value component" in this paper is mainly focused on the C4 dataset (Figures 1 and 3). What are the similarities and differences between the analysis on other datasets and the C4 dataset? Especially on datasets that are very different from the C4 dataset. In addition, these analyses are mainly based on 16 randomly sampled training samples. When the number of samples increases or decreases, what changes will occur in the analysis results?
- As shown in Figure 3, there is a clear difference in the importance of different linear layers (such as self_attn.k_proj vs. self_attn.up_proj), what this mainly stems from, the authors can give more comments on this.
- What is the relationship between the sparsity ratio in the proposed R-Sparse and the final inference acceleration? For example, what is the corresponding acceleration for a certain sparsity ratio?
- In Figure 6, why under Dense, when the Generation Length becomes longer (1024->2048), the generation speed slows down, while when it is 128->256->512, the generation speed is accelerated.

**Questions:**

See the Weaknesses section

---

> ### Author Response · Authors · 2024-11-22
> **Response to Reviewer zFFb**
>
> We sincerely thank Reviewer zFFb for the positive feedback and constructive suggestions. To address Reviewer zFFb’s concerns, we provide point-by-point responses below.
>
> **[Q1: Importance observation across different datasets and varying number of samples.]**: Thanks for the question. We conducted additional experiments to assess the contributions of each input channel and singular value components across various datasets and different numbers of samples. The results are detailed in Section B.2 of the updated draft. Our findings consistently show that the primary contributions of importance are concentrated in the lower-right corner of the matrix. This supports the use of R-Sparse for accelerating inference without losing performance.
>
> **Details:** To validate these observations across datasets, we examined diverse domains from the RedPajama dataset, including ArXiv, GitHub, StackExchange, and the original C4 dataset, ensuring a wide range of data diversity. Additionally, we evaluated varying sample sizes, ranging from 1 to 1024, and observed consistent patterns across all configurations.
>
>
> **[Q2: Explanation of varying importance observed across linear layers.]**: Thanks for the suggestion. The varying importance mainly comes from the intrinsic low-rank properties of different linear layers. When the linear layer are more low-rank, the importance tend to centrarate only on the top-singular value components, results in a more sparse patterns in Figure 3 (i.e., the self_attn.k_proj) while for relatively higher rank, the importance patterns tend to distribute more uniformly across the vertical axis, (e.g., the mlp.up_proj). The layerwise low-rank properties aligns with previous investigations, such as Figure 4 in [1] and related with the learning dynamics of transformers(e.g., Section 4 in [2]).
>
> [1] From GaLore to WeLore: How Low-Rank Weights Non-uniformly Emerge from Low-Rank Gradients.
>
> [2] JoMA: Demystifying Multilayer Transformers via JOint Dynamics of MLP and Attention.
>
> **[Q3: Relationship between sparsity ratio and final inference acceleration.]**: Thanks for the question. Ideally, the acceleration equals to $\frac{1}{1-s}$ where s is the sparsity ratio. That is, 50% sparsity theoretically allows for a 2x speedup. However, in practice, due to the detailed memory access pattern, such as memory blocks not aligning perfectly with rows of the weight matrix, there is overhead associated with processing zero values. Additionally, activation sparsity doesn’t change the complexity of the scaled dot product in attention, as well as the KV cache overhead.  Consequently, our implementation achieves a 1.4x speedup for 50% sparsity. For a more detailed analysis, we evaluate the
> latency of a single MLP block across different sparsity ratios. As shown in Table R1, the acceleration is gradually improved as the sparsity ratio increases. We evaluate on a single MLP block with input dimension of 4096 and hidden dimension of 11008, the latency is averaged by 80 runs after warm up running.
>
> Table R1: Comparison of the latency across different sparsity ratios.
>
> | Sparsity Ratio  | 0      | 0.1    | 0.3    | 0.5    | 0.7    | 0.9    |
> | ------------------ | ------ | ------ | ------ | ------ | ------ | ------ |
> | Latency (ms)       | 0.8190 | 0.7386 | 0.5792 | 0.4172 | 0.2567 | 0.1233 |
> | Reduction (%)      | 0      | 0.0982 | 0.2928 | 0.4906 | 0.6866 | 0.8495 |
>
> **[Q4: Explanation of generation speed in Figure 6.]**: Thanks for the question. For dense inference, the generation latency is primarily composed of three components: (i) the latency of prefilling, denoted as $t_{prefill}$, (ii) (decoding phase) the latency for computation and memory access of model parameters, $t_{weight}$, and (iii) (decoding phase) the latency for memory access of the KV cache, $t_{kv}(n_{prompt} + n_{generation})$ that linearly grows with the sequence length ($n_{prompt} + n_{generation}$). Therefore, the reported generation speed can be expressed as: $\frac{t_{prefill} + n_{generation}t_{weight}+n_{generation}t_{kv}(n_{prompt} + n_{generation})}{n_{generation}}$. In the early stages of decoding, the memory overhead of the KV cache is relatively small compared to that of the model parameters. As a result, the $\frac{t_{prefill}}{n_{generation}}$ term has significant impact on the end-to-end latency, but its effect diminishes as $n_{generation}$ increases (For generation length from 128 to 512), while for longer sequence length, the memory access cost of KV cache becomes more significant, as its size grows with the sequence length, leading to a reduction in the end-to-end generation speed.

---

> > ### Author Response · Authors · 2024-11-25
> > **We are keen to discuss further with you**
> >
> > Dear Reviewer zFFb,
> >
> > We sincerely appreciate the time and effort you have dedicated to reviewing our work. Your insightful suggestions have been invaluable in improving the quality of our work. We have carefully addressed each of your concerns and made the necessary revisions in the updated draft.
> >
> > As the discussion period deadline approaches, we would be grateful if you could review our responses and let us know if there are any additional questions. Thank!
> >
> > Best,
> >
> > Authors

---

> > > ### Comment · Reviewer_zFFb · 2024-11-26
> > >
> > > Thank you for the detailed responses. My concerns have been addressed and I maintain my positive score.

---

### Official Review · Reviewer_T8xk · 2024-11-04

**Soundness:** 3
**Presentation:** 2
**Contribution:** 2
**Rating:** 5
**Confidence:** 3

**Summary:**

This paper presents R-Sparse, a training-free activation sparsity approach for large language models (LLMs). Current activation sparsity methods face limitations with non-ReLU activation functions and have difficulties in predicting active channels and achieving high sparsity ratios. R-Sparse overcomes these challenges by leveraging the sparsity of input channels and singular value components. The authors conduct investigations and find that non-sparse components can be regarded as bias terms and full computation can be approximated by a combination of input channels and weight singular values. R-Sparse is applied to both attention and MLP modules of LLMs and an evolutionary search algorithm is used to find optimal sparse component ratios. Experiments on Llama-2/3 and Mistral models across ten tasks show that R-Sparse achieves 50% model-level sparsity with comparable performance and up to 43% end-to-end speed improvement with a customized kernel.

**Strengths:**

* The training-free method does not require extensive pre-training, making it more efficient and easier to implement compared to methods that need continual training.
* R-Sparse achieves high sparsity levels (50% model-level sparsity) without sacrificing performance, leading to significant improvements in efficiency.
* R-Sparse is compatible with weight quantization for further efficiency gains and can be applied to different LLM families and a variety of tasks.

**Weaknesses:**

* Table 1 is difficult to interpret. From the description of the authors, R-Sparse40% is compared with CATS22% and GRIFFIN33%, if my understanding is correct. Therefore, R-Sparse does not consistently outperform CATS across all tasks (e.g., PIQA 78.24 vs 79.00). The authors are suggested to refine the claim to avoid the misleading. Meanwhile, for certain cases, the performance increases with an even higher sparsity ratio (e.g., 79.49 vs 79.92 for R-Sparse40% and R-Sparse50%) on PIQA. Could the authors provide some insights into this phenomenon?
* The sensitivity analysis of hyperparameters should be added for a more thorough investigation of the effectiveness of R-Sparse.
* As [1] is discussed by the authors in the related works, why do the authors choose not to compare with [1] in the experiments?
* The authors are suggested to include the complexity analysis and running time comparison of R-Sparse, especially regarding the evolutionary search algorithm.
* The writing can be further improved. For example, the optimal sparse-rank $\alpha$ is not formally defined. From Algorithm 1, $\alpha$ seems to be fixed, how is it optimized?

[1] Deja Vu: Contextual Sparsity for Efficient LLMs at Inference Time, ICML 2023

**Questions:**

Please kindly refer to the Weaknesses.

---

> ### Author Response · Authors · 2024-11-22
> **Responses to Reviewer T8xk (Q1-Q4)**
>
> We thank Reviewer T8xk for the valuable questions and suggestions. We elaborate more details of our method and provide point-wise responses in the following:
>
> **[Q1: Explanation of Table 1]** Thanks for the question. R-Sparse 40% is not compared with CATS 22% and GRIFFIN 33%. Instead, we compare R-Sparse with other baselines under the same sparsity ratios (As discussed in Line 392), that is R-Sparse 40% v.s. CATS 40% and R-Sparse 50% v.s. GRIFFIN 50\% where R-Sparse consistently outperforms both CATS and GRIFFIN.  Additionally, we choose sparsity ratio of 22% for CATS and 33% for GRIFFIN for the reason that is the sparsity reported by the original publications where 50% mlp-level sparsity are applied and CATS only sparsifies mlp.gate and mlp.down while GRIFFIN sparsify the whole mlp modules. We’ve included this sparsity selection details in Line 370. And We’ve adjusted the lines in Table 1 in the updated draft to avoid misleading where R-Sparse 40% and CATS 40% are placed in adjacent lines as well as R-Sparse 50% and GRIFFIN 50%.
>
> Additionally, we report the performance of various methods across different sparsity ratios in Figure 5, that provides a more thorough comparison and demonstrates our method consistently outperforms other baselines by a clear margin.
>
> In certain cases, higher sparsity leads to improved performance. That’s because appropriate sparsity may help mitigate overfitting and enhance model generalization. This phenomenon has been demonstrated in lots of prior studies. Such as Figure 1 in SparseGPT[1], Figure 2 in Essential Sparsity[2] and Figure 4 in GRIFFIN[3].
>
> [1] SparseGPT: Massive Language Models Can be Accurately Pruned in One-Shot.
>
> [2] The Emergence of Essential Sparsity in Large Pre-trained Models: The Weights that Matter.
>
> [3]  Prompt-prompted Adaptive Structured Pruning for
> Efficient LLM Generation.
>
>
> **[Q2: Sensitivity analysis of hyperparameters]**: Thank you for pointing this out. We conducted a sensitivity analysis of the hyperparameters in Section 4.4 (A2 and A3), focusing specifically on the sparse-rank ratio $\rho$ for each linear layer. As discussed in Section 4.4 (A2), we compared R-Sparse with its sparse ($\rho=1$) and low-rank ($\rho=0$) counterparts, observing significant improvements, as presented in Table 3. In Section 4.4 (A3), we further compared the layer-wise $\rho$ values obtained through Algorithm 1 and the corresponding uniform $\rho$ assignment. As shown in Table 4, the adaptive $\rho$ consistently outperformed the uniform assignment across various sparsity ratios and tasks. These ablation studies provide a comprehensive evaluation of the effectiveness of R-Sparse.
>
>
> **[Q3: Comparison with Deja Vu]**: Thanks for the question. Deja Vu requires training an additional predictor to identify active channels, whereas our comparison focuses exclusively on training-free approaches. To further address Reviewer T8xk’s concern, we conducted a comparison with Deja Vu using an oracle predictor—i.e., a predictor that perfectly identifies active channels with 100% accuracy. The results, presented in Table R2, demonstrate that our method outperforms Deja Vu (Oracle), achieving an average accuracy improvement of 0.51. Note that in realistic settings, the inaccuracy of Deja Vu’s predictor would further degrade its performance.
>
> Table R2: Comparison of R-Sparse with Deja Vu (Oracle) under 50% sparsity on Llama-2-7B.
>
> | Method           | WG    | PIQA  | SciQ  | OBQA  | HS    | BoolQ | Arc-E | Arc-C | Average |
> | ---------------- | ----- | ----- | ----- | ----- | ----- | ----- | ----- | ----- | ------- |
> | Llama-2-7B       | 69.14 | 78.07 | 93.80 | 31.40 | 57.13 | 77.71 | 76.35 | 43.43 | 65.88   |
> | R-Sparse         | 67.40 | 77.31 | 93.90 | 31.40 | 54.26 | 72.84 | 74.58 | 40.78 | 64.06   |
> | Deja Vu (Oracle) | 67.09 | 76.66 | 92.80 | 29.80 | 55.14 | 72.87 | 73.40 | 40.61 | 63.55   |
>
>
> **[Q4: Complexity analysis and running time comparison of R-Sparse]**: Good suggestions. R-Sparse is a training-free approach, with its primary computational overhead stemming from the evolutionary search algorithm. The time cost of this search is linearly proportional to the population size ($P$), the total number of generations ($G$), and the number of samples used for perplexity evaluation ($N$), resulting in a complexity of $O(PGN)$. As discussed in Line 322, our experiments use a population size of 32, a generation count of 5, and 16 samples for perplexity evaluation. In a specific case, the search process requires approximately one hour on a single A6000 GPU for the Llama-2-7B model, which is negligible compared to the training-level computational cost.

---

> ### Author Response · Authors · 2024-11-22
> **Responses to Reviewer T8xk (Q5)**
>
> **[Q5: Writing improvement]**: Good suggestions. The optimal sparse-rank ratio is defined as $\rho^* = \mathrm{argmin}{\rho} \mathcal{L}(f, \rho)$ where the loss function $\mathcal{L}$, represents the average perplexity computed over 16 randomly selected samples from the C4 training set, and $f$ denotes the original LLMs. To solve this optimization problem, we employ an evolutionary search algorithm, as outlined in Algorithm 1 of the PDF. In Algorithm 1, the variable $\alpha$ is a random variable used to control the crossover of individual solutions. It does not require optimization; instead, the sparse rank ratio $\rho$ will be optimized during each generation as shown in Line 10.

---

> > ### Comment · Reviewer_T8xk · 2024-11-25
> > **Thank you for the response**
> >
> > I would like to thank the authors for the replies, which solve most of my concerns. One remaining concern is that by referring to the sensitivity analysis of hyperparameters, I mean the hyperparameters used in the evolutionary search algorithm (i.e., use a population size of 32, a generation count of 5, and 16 samples for perplexity evaluation). Is the choice heuristic or optimized within the search space?

---

> > > ### Author Response · Authors · 2024-11-25
> > > **Thank you for the follow-up question**
> > >
> > > Thanks for the follow-up question. These hyperparameters are heuristically selected based on some preliminary experiments. Specifically: (i) The population size primarily controls the diversity of individual solutions. A larger population reduces the risk of premature convergence to local optima but slows down the convergence speed. In our preliminary experiments, we tested population sizes of  $\{8, 16, 32, 128\}$ and observed final perplexity values of \{9.85, 7.15, 6.32, 6.33\}, respectively. Based on these results, we selected a population size of 32, as increasing it further provided no significant improvement. (ii) Then, we evaluated a generation count of 10 and found that convergence is nearly stable by generation 5, as shown in Table R2. Thus, we chose a generation count of 5. (iii) Note that the time cost for evolutionary search grows linearly with the number of samples used. To balance the trade-off between search quality and practical time constraints, we selected 16 samples to evaluate each individual solution. And this configuration (32 population size, 5 generation count and 16 samples) takes approximately one hour of overhead on a single A6000 GPU, that is far less than training-level time cost.
> > >
> > > Table R2: Best perplexity achieved at different generation steps
> > >
> > > | Generation Step          | 1    | 2  | 3  | 4  | 5    | 6 | 7 | 8 | 9 | 10 |
> > > | ---------------- | ----- | ----- | ----- | ----- | ----- | ----- | ----- | ----- | ------- |------- |
> > > |Llama-2-7B|13.58|12.42|11.44|7.85|6.32|6.32|6.32|6.20|6.20|6.19|

---

> > > > ### Author Response · Authors · 2024-11-26
> > > > **We are keen to discuss further with you**
> > > >
> > > > We sincerely thank Reviewer T8xk for their efforts throughout the review and discussion period. We have provided a detailed explanation of the hyperparameter choices in evolutionary search, supported by several preliminary ablation studies. We would greatly appreciate it if reviewer 78xk could review our responses and let us know if there are additional questions. Thanks

---

> ### Comment · Reviewer_T8xk · 2024-11-27
>
> Thank you for the additional results. I have no further questions and will increase my rating accordingly.

---

> > ### Author Response · Authors · 2024-11-27
> > **Thanks for responses**
> >
> > Thank you for your responses. We’re glad that our responses addressed your concerns. However, the adjusted rating suggests our work is not yet suitable for acceptance. Could you kindly provide further suggestions or questions for improvement so that we can enhance our work to meet the required standards for acceptance? Thank you!

---

### Comment · Area_Chair_qRzt · 2024-12-03
**End of reviewer-author discussion phase**

Dear reviewers,

As we near the conclusion of the reviewer-author discussion phase, I wanted to kindly follow up to see if you’ve had a chance to review the author responses on your comments. Could you confirm that you’ve read it and, if needed, update your review and scores accordingly?

Thank you for your time and effort!

Your AC

---

### Meta-Review · Area_Chair_qRzt · 2024-12-22

**Metareview:**

a) Scientific Claims and Findings:
The paper introduces R-Sparse, a training-free activation sparsity method for efficient LLM inference. Key findings show that non-sparse input components can be treated as bias terms, and computation can be approximated using input channels and weight singular values. The method achieves 50% model-level sparsity while maintaining performance, with claimed 43% efficiency improvements using customized kernels.

(b) Strengths:
The paper presents a novel training-free approach that achieves high sparsity levels while maintaining performance. It's broadly applicable to both attention and MLP modules, works across different LLM architectures, and is compatible with weight quantization. The experimental validation is comprehensive across diverse tasks.

(c) Weaknesses:
The paper has some presentation issues around key definitions and concepts. There are questions about hyperparameter sensitivity and incomplete comparisons with some baselines. The relationship between sparsity ratio and actual speedup needs better clarification. I recommend the authors to work on these for the camera ready version.

(d) Reasons for Acceptance:
The paper warrants acceptance based on three key factors:
It presents a novel technical solution to an important problem in LLM inference optimization, achieving significant sparsity while maintaining performance.
The method has broad practical applicability across different architectures and modules.
The experimental validation is thorough, with clear improvements demonstrated across multiple tasks.

While there are some presentation issues, these can be addressed in the camera-ready version. The core technical contribution, novelty of of viewpoint to sparsity, and practical utility make this paper a valuable addition to the field.

**Additional Comments On Reviewer Discussion:**

The review process featured extensive and constructive discussion between authors and reviewers. Throughout the discussion period, the authors demonstrated strong engagement and responsiveness to reviewer concerns. Reviewer T8xk initially raised questions about result interpretation and methodology, to which the authors provided additional experiments and clarifications that addressed most concerns. Reviewer zFFb's questions about dataset analysis and architectural variations were thoroughly addressed with additional experiments, leading to a positive recommendation. While Reviewer LhWB raised important points about clarity and definitions, the authors made earnest efforts to clarify these issues, and the remaining concerns are primarily about presentation rather than technical substance. Reviewer sWLw's initial concerns about novelty were effectively addressed through detailed responses and additional experimental validation.
The authors' thorough responses and willingness to conduct additional experiments demonstrate their commitment to scientific rigor. While some presentation issues remain, these can be addressed in the final version. The positive recommendations from multiple reviewers, combined with novel viewpoint to sparsity and efficiency and broad practical applicability, support accepting this paper for publication at ICLR 2025.

---

### Decision · Program_Chairs · 2025-01-22

Accept (Poster)